

# Incidence rate, risk factors, and management of Bell's palsy in the Qurayyat region of Saudi Arabia

Fahad Alanazi[1], Faizan Z. Kashoo[2], Anas Alduhishy[3], Mishal Aldaihan[4], Fuzail Ahmad[2] and Ahmad Alanazi[2]

[1] Department of Physical Therapy and Rehabilitation Sciences, College of Applied Medical Sciences, Jouf University, Al Jouf, Saudi Arabia
[2] Department of Physical Therapy and Health Rehabilitation, College of Applied Medical Sciences, Majmaah University, Al Majmaah, Riyadh, Saudi Arabia
[3] Department of Physical Therapy, College of Applied Medical Sciences, Imam Abdulrahman Bin Faisal University, Dammam, Saudi Arabia
[4] Department of Rehabilitation Sciences, College of Applied Medical Sciences, King Saud University, Riyadh, Saudi Arabia

## ABSTRACT

**Background.** Bell's palsy is an idiopathic facial nerve dysfunction causing temporary paralysis of muscles of facial expression. This study aimed to determine the incidence rate, common risk factors, and preferred treatment by the Saudi patients with Bell's palsy.

**Method.** This cross-sectional study was carried out in the Qurayyat region of Saudi Arabia. The retrospective medical records were searched from 2015–2020 of patients diagnosed with Bell's palsy at Qurayyat General Hospital and King Fahad hospital. A 28-item questionnaire was developed by a team of experts and pre-tested among patients with Bell's palsy before being sent to the eligible participants. The data were analyzed using summary statistics, Chi-square test, Fisher exact test and Likelihood ratio test.

**Results.** We identified 279 cases of Bell's palsy from the medical records of the hospitals from the years 2015 to 2020, accounting for 46.5 cases per year and an incidence rate of 25.7 per 100,000 per year. Out of 279 patients with Bell's palsy, only 171 returned the questionnaire accounting for a response rate of 61.2%. Out of 171 patients with Bell's palsy, females ($n = 147$, 86.0%) accounted for the majority of cases. The most affected age group among participants with Bell's palsy was 21–30 years ($n = 76$, 44.4%). There were 153 (89.5%) cases who reported Bell's palsy for the first time. The majority of the participants experienced right-sided facial paralysis ($n = 96$, 56.1%). Likelihood ratio test revealed significant relationship between exposure to cold air and common cold with age groups ($\chi^2(6, N = 171) = 14.92$, $p = 0.021$), $\chi^2(6, N = 171) = 16.35$, pp $= 0.012$ respectively. The *post hoc* analyses revealed that participants in the age group of 20–31-years were mostly affected due to exposure to cold air and common cold than the other age groups. The main therapeutic approach preferred was physiotherapy ($n = 149$, 87.1%), followed by corticosteroids and antivirals medications ($n = 61$, 35.7%), acupressure ($n = 35$, 20.5%), traditional Saudi herb medicine ($n = 32$, 18.7%), cauterization by hot iron rod ($n = 23$, 13.5%), supplementary therapy ($n = 2$, 1.2%), facial cosmetic surgery ($n = 1$, 0.6%) and no

Corresponding author
Faizan Z. Kashoo,
f.kashoo@mu.edu.sa

treatment ($n = 1$, 0.6%). The most preferred combined therapy was physiotherapy (87.6%) with corticosteroid and antiviral drugs (35.9%), and acupressure (17.6%).
**Conclusion**. The rate of Bell's palsy was approximately 25.7 per 100,000 per year in the Qurayyat region of Saudi Arabia. Exposure to cold air and common cold were the significant risk factors associated with Bell's palsy. Females were predominantly affected by Bell's palsy in the Qurayyat region of Saudi Arabia. Bell's palsy most commonly occurred in the age group 21–30 years. The most favored treatment was physiotherapy following Bell's palsy.

# INTRODUCTION

Bell's palsy is a common lower motor nerve paralysis of the facial nerve of unknown origin (*Eviston et al., 2015*). The patient with Bell's palsy experiences sudden unilateral (rarely bilateral) flaccid paralysis of muscles that control facial expression (*Gilden, 2004*). The patient is unable to perform facial movements on the affected side, and facial asymmetry is clearly shown with attempted facial movement (*Reich, 2017*).

The annual incidence rate (1992–1996) was reportedly 20.2 per 100,000 population, according to the UK General Practice Research database (*Rowlands et al., 2002*). Research studies globally have reported variations in the annual incidence rate (11–50 cases per 100,000) of Bell's palsy (*Hsieh, Wang & Lee, 2013*; *JI, Prim-Espada & Fernández-García, 2005*; *Kokotis & Katsavos, 2015*; *Monini et al., 2010*; *Yilmaz et al., 2019*). The national prevalence of Bell's palsy in Saudi Arabia is unknown; however, regional incidence/prevalence has been reported in a few studies, such as 5.3 per 100,000 per year (1992–1995) in the Asir region (*Al Ghamdi, 1997*), 30.4 cases per 100,000 per year (1995–1997) in the Qassim region (*Hamid, 1998*), 26.3–27.8 cases per 100,000 per year (2011–2012) in the Aljouf region (*Jamil et al., 2013*), and 26.3 cases per 100,000 per year (2016–2017) in the Arar region (*Alanazi et al., 2017*).

The cause of Bell's palsy is idiopathic; however many possible causes have been recognized, such as reactivation of the herpes simplex virus, human immunodeficiency virus, and hepatitis C virus (*Greco et al., 2012*). Additionally, there are several risk factors associated with Bell's palsy, including age, pregnancy, epilepsy, obesity, hypertension, diabetes, respiratory tract infection, vaccination (*Colella, Orlandi & Cirillo, 2021*; *Potterton, 2015*), and genetic susceptibility due to consanguineous marriages in Saudi Arabia (*Middle et al., 2007*). The recent increase in the prevalence of diabetes (*Elhadd, Al-Amoudi & Alzahrani, 2007*), hypertension (*Al-Nozha et al., 2007*), and obesity (*Al-Nozha et al., 2005*) in Saudi Arabia increase the risk of Bell's palsy. In addition, the customary practice of consanguineous marriage increases the risk of autosomal recessive genetic disorders (*AbdulAzeez et al., 2019*). Therefore, a study is needed to explore the possible impact of

increased risk factors and consanguineous marriage on the incidence rate of Bell's palsy in the Saudi population.

The standard recommended treatment of Bell's palsy is oral corticosteroids, with limited evidence concerning the additive benefit of antiviral drugs for 10–12 days (*Allen & Dunn, 2004*; *Engström et al., 2008*; *Heckmann et al., 2019*; *Salinas et al., 2010*). In addition, physiotherapy (*Gatidou et al., 2021*), acupuncture (*Liang et al., 2006*), dry needling (*Zhang, Wang & He, 2007*), taping (*Ghous et al., 2018*), and neural mobilization techniques (*Kashoo, Alqahtani & Ahmad, 2021*) have been reported to be beneficial. However, traditional methods of treating Bell's palsy in Saudi Arabia are still in practice. Two of the common traditional methods for general pain management are the use of herbs (*Shaikh, 2021*) and cauterization (*Aboushanab & AlSanad, 2019*). Cauterization for Bell's palsy is performed by placing a hot iron rod on the back of the neck or between the thumb and index finger (*Alsanad et al., 2018*). Such traditional methods may result in severe burn injuries and sometimes develop into complicated wounds in cases with pre-existing conditions, such as diabetes (*Qureshi et al., 2018*).

There is limited literature on the contribution of risk factors and the treatments preferred by the Saudi population following a diagnosis of Bell's palsy. Therefore, the aim of this study is to determine the incidence rate, possible risk factors, and preferred treatment options for Bell's palsy in the Saudi population.

## MATERIALS & METHODS

### Study design and setting

The study was a retrospective cross-sectional hospital-based study. The study was carried out in the department of physiotherapy at the Qurayyat General Hospital and King Fahad Hospital. Ethical approval was obtained from the ethical committee of the hospitals in December 2020 prior to the commencement of the study (QGH-EC-16-2020).

The files and medical records of patients diagnosed with Bell's palsy in Qurayyat General Hospital and King Fahad Hospital were reviewed. A total of 279 patients from the years 2015 through 2020, inclusively, were identified and contacted by email and telephone for participation. If the patient was a minor, the parents were contacted for consent (Fig. 1). The subjects were included in the study if they had been diagnosed with Bell's palsy by a qualified medical doctor. One subject who was recruited through a community advertisement was assessed by a neurologist at Qurayyat General Hospital.

### Population and sample

The target population was Bell's palsy patients in the Qurayyat region of Saudi Arabia. The sample size from reviewing medical records was 279. To achieve a 95% confidence interval with a 5% margin of error and 50% response distribution, the current study required 162 Bell's palsy patients to represent the population (http://www.raosoft.com/samplesize.html). Out of 279 eligible participants, 171 participants with Bell's palsy responded to the questionnaire, representing a 61.2% response rate.

 

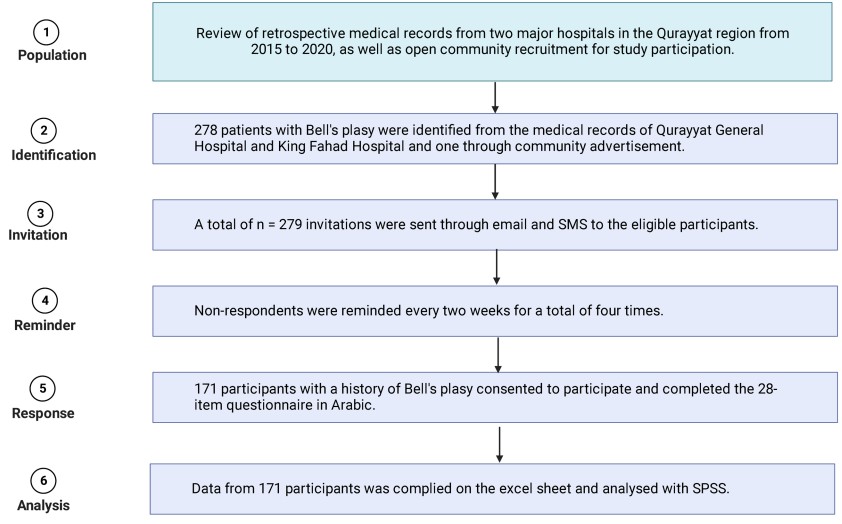

**Figure 1 Flow chart of invitation, recruitment and response rate.**

## Questionnaire

A group of experts consisted of four clinicians (two neurologists, two general physicians) and four academic university staff (one associate professor in physiotherapy, one language expert, two professors in medical college) with an average experience of more than 10 years. A preliminary set of questions was submitted by the authors of this study to the expert committee. The preliminary set of questions was emailed to every member of the expert committee before meeting. The expert committee conducted two meetings before finalizing a set of 28 questions (Appendix 1). Out of 28 questions in the questionnaire, eight multiple-choice questions were related to gender, age, onset, recurrence, number of previous episodes of Bell's palsy, side involved, and smoking. Sixteen dichotomous questions related to a previous history of stroke, neurological disorders, cardiovascular disorders, respiratory diseases, birth defects, eye defects, balance problems, viral infection, common cold, flu, high blood pressure, diabetes, ear infection, head injury, previous surgery, and COVID-19 vaccination. Three dichotomous questions related to genetics, such as consanguinity, family history of Bell's palsy, and genetic disorders. The last one was an open-ended question related to the selected treatment for Bell's palsy.

To evaluate the questionnaire for clarity, it was pre-tested among 10 Bell's palsy patients visiting Qurayyat General Hospital. Any question identified as unclear by patients was rephrased by the expert committee until all expert members approved the changes.

The final version of the questionnaire was piloted among 30 patients with Bell's palsy visiting Qurayyat General Hospital and King Fahad Hospital.

## Procedure

The retrospective medical records of two major hospitals (Qurayyat General Hospital and King Fahad Hospital) were searched for patients diagnosed with Bell's palsy. Contact was

made with eligible patients by SMS, telephone, and email. An advertisement to participate was circulated in hospitals and community centers, such as shopping centers, in the form of pamphlets. All eligible participants were invited to participate in the study. Those who consented were asked to fill out an online questionnaire consisting of 28 questions. Non-respondents were contacted again after an interval of two weeks up to a maximum of four times until all communications were stopped.

## Statistical analysis

The information and data from the study were entered into an electronic database (SPSS® for Windows®V.20). The demographic data were analyzed through frequency distribution, and the relation between various risk factors was analyzed by a chi-square test, Fishers exact test, and likelihood-ratio test. The total population of the Qurayyat region of Saudi Arabia in the year 2020 was 180,430, according to the Ministry of Health (MOH), Saudi Arabia (*MOH, 2018*). The incidence rate was calculated as the number of new Bell's palsy cases that appeared annually per 100,000 population. The incidence rate was calculated by dividing the total number of cases identified ($n = 279$) by cases with the total population ($n = 180,430$) at the region per 100,000 per year.

## RESULTS

Among the 171 Bell's palsy patients participating in the study, most were female ($n = 147$, 86%). The highest number of patients with Bell's palsy was among the group aged 21–30 years ($n = 76,44.4\%$), and the lowest was among those aged 1–10 years ($n = 11,6.4\%$). Most were reporting Bell's palsy for the first time ($n = 129$, 75.45%), and 10.5% ($n = 18$) were reporting recurrent Bell's palsy. A total of 21 (12.3%) participants had been vaccinated with COVID-19 before experiencing Bell's palsy (Table 1).

A significant number of participants ($n = 135,78.9\%$) reported being exposed to cold air before experiencing Bell's palsy. A likelihood-ratio test was performed to examine the relation between age group and exposure to cold air before experiencing Bell's palsy and was found to be significant, $\chi^2(5, n = 171) = 14.92$, $p = 0.011$. The *post hoc* analysis with Bonferroni correction and adjusted *p*-value of 0.007 was found to be significant at the p < 0.05 level and revealed that those aged 21–30 years were significantly affected. Sixty-five (38%) participants reported that their parents were cousins, and 20 (11.7%) reported having a familial genetic disorder. There was no significant relationship between consanguinity with gender, onset, and recurrence of Bell's palsy; however, consanguinity showed a significant relationship with side affected and age group $\chi^2(2, N = 171) = 12.090$, $p = 0.002$; $\chi^2(5, N = 171) = 13.025$, $p = 0.023$ respectively.

The main preferred therapeutic approach was physiotherapy ($n = 149$, 87.1%), followed by corticosteroids and antiviral drugs ($n = 61$, 35.7%), acupressure ($n = 35$, 20.5%), traditional Saudi herb medicine ($n = 32$, 18.7%), cauterization by hot iron rod ($n = 23$, 13.5%), supplementary therapy (vitamins and neuro-vitality drugs ($n = 2$, 1.2%), facial cosmetic surgery ($n = 1$, 0.6%), and no treatment ($n = 1$, 0.6%) (Table 2).

Relatively fewer participants had suffered from ear infections ($n = 28$, 16.4%), diabetes ($n = 23$, 13.5%), genetic diseases ($n = 20$, 11.7%), high blood pressure ($n = 18$, 10.5%),

**Table 1 Demographic characteristics of participants.**

| Variables | Number (n = 171) | Percentage (%) | P |
|---|---|---|---|
| **Gender** | | | |
| Male | 24 | 14 | |
| Female | 147 | 86 | 0.001[*] |
| **Age group (years)** | | | |
| 1–10 | 11 | 6.4 | |
| 11–20 | 32 | 18.7 | |
| 21–30 | 76 | 44.4 | |
| 31–40 | 16 | 9.4 | 0.001[**] |
| 41–50 | 20 | 11.7 | |
| 51-above | 14 | 8.2 | |
| **Side affected** | | | |
| Right | 96 | 56.1 | |
| Left | 63 | 36.8 | 0.001[**] |
| Bilateral | 12 | 7 | |
| **Onset** | | | |
| Sudden | 129 | 75.4 | |
| Gradual | 42 | 24.6 | 0.001[*] |
| **Recurrence** | | | |
| First time | 153 | 89.5 | |
| Second time or more | 4 | 2.3 | 0.001[*] |
| **Treatment following Bell's palsy** | | | |
| Physical therapy | 149 | 87.1 | |
| Traditional | 32 | 18.7 | |
| Drugs | 61 | 35.7 | |
| Acupressure | 35 | 20.5 | 0.001[***] |
| Hot iron | 23 | 13.5 | |
| Others | 4 | 2.3 | |
| **COVID-19 vaccination** | | | |
| Before vaccination | 150 | 87.7 | |
| After vaccination | 21 | 12.3 | 0.001[*] |

Notes.
[*] One sample binomial test.
[**] One sample chi-square test.
[***] Fisher's exact test.

neurological disorder ($n = 16$, 9.4%), head injury ($n = 11$, 6.4%), balance problems ($n = 10$, 5.8%) stroke ($n = 3$, 1.8%), or heart disease ($n = 3$, 1.8%) (Fig. 2).

# DISCUSSION

This study aimed to determine the incidence rate, risk factors, and preferred treatment among participants with Bell's palsy residing in the Qurayyat region of Saudi Arabia. According to this study, females were predominantly affected, and a significant number of participants opted for physiotherapy and corticosteroid and antiviral drug therapy.

Table 2 Mutually inclusive responses from participants about preferred treatment following Bell's palsy.

| Preferred treatment | Physical therapy | Hot iron | Allopathic drugs | no treatment | Acupressure | Cosmetic surgery | Supplements | Hot iron | Mutually inclusive responses |
|---|---|---|---|---|---|---|---|---|---|
| | N | N | N | N | N | N | N | N | |
| Physical therapy | 149 | 22 | 57 | 0 | 30 | 1 | 2 | 15 | 429 |
| Allopathic drugs | 57 | 9 | 61 | 0 | 15 | 0 | 0 | 6 | 231 |
| Hot iron | 22 | 32 | 9 | 0 | 5 | 0 | 1 | 5 | 114 |
| No treatment | 0 | 0 | 0 | 1 | 1 | 0 | 0 | 0 | 3 |
| Acupressure | 30 | 5 | 15 | 1 | 35 | 1 | 0 | 6 | 144 |
| Cosmetic surgery | 1 | 0 | 0 | 0 | 1 | 1 | 0 | 0 | 5 |
| Supplements | 2 | 1 | 0 | 0 | 0 | 0 | 2 | 0 | 8 |
| Hot iron | 15 | 5 | 6 | 0 | 6 | 0 | 0 | 23 | 74 |
| Total | 149 | 32 | 61 | 1 | 35 | 1 | 2 | 23 | 468 |

Notes.
  N, number of responses from patients with Bell's Palsy.

The average incidence rate of Bell's palsy was found to be 25.7 cases per 100,000 per year in the Qurayyat region of Saudi Arabia. The most affected age group comprised those aged 21–30 years, with females being 6.12 times more likely to be affected than males. Physiotherapy and standard drug therapy (corticosteroid and antiviral drugs) were preferred over the other modes of treatment. A study conducted in the Arar region of Saudi Arabia found 26.3% of cases of Bell's palsy with females (61%) more affected than males (*Alanazi et al., 2017*) and that participants preferred physiotherapy treatment over drug therapy (*Alanazi et al., 2017*).

Most (75%) of the participants with Bell's palsy experienced sudden facial muscle paralysis, with the majority having experienced first-time facial paralysis and 10.5% reporting recurrent facial paralysis. A study conducted in the Asir region of Saudi Arabia Bell's reported a 5.35 per 100,000 population per year incidence rate of Bell's palsy, with most patients reporting sudden onset Bell's palsy in winter (*Al Ghamdi, 1997*). In contrast, the incidence rate of Bell's palsy in our study was approximately 25.7 per 100,000 per year. The incidence rate was calculated based on the total population of the Qurayyat region, which was 180,430, according to the 2020 population census (*Ministry of Health Saudi Arabia, 2020*).

In our study, we found that 12.3% of participants reported Bell's palsy after the COVID-19 vaccination. Research on the association of Bell's palsy with the COVID-19 vaccination is scarce, with few case reports having been presented. Some studies have reported a significant association between COVID-19 vaccination and the incidence rate of Bell's palsy (*Cirillo & Doan, 2021*); however, it is possible that these studies introduced selection
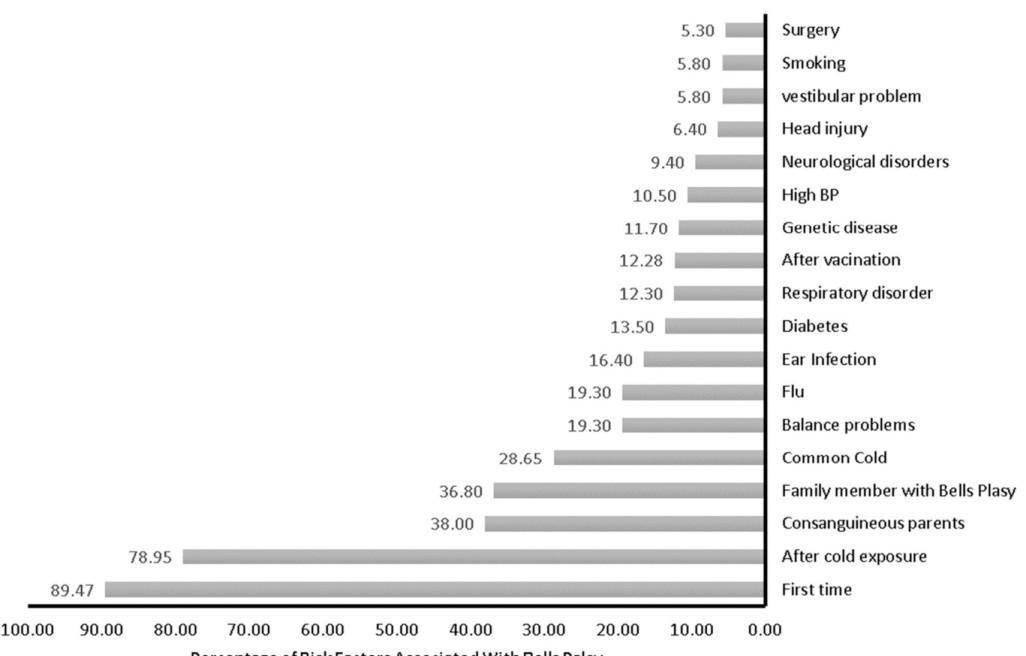

**Figure 2  Risk factors associated with Bell's palsy.**

bias in terms of age groups vaccinated, as the incidence rate of Bell's palsy varies greatly with age (*Li et al., 2021*). In our study, we found that 78.9% of participants reported Bell's palsy after being exposed to cold air. However, only 28.7% and 19.3% of participants reported having had a common cold or flu before suffering from Bell's palsy. A study conducted among 1,181 active duty military service members in the United States reported a 33% higher incidence rate of Bell's palsy in cold climates than in warm regions (*Campbell & Brundage, 2002*).

Pre-existing conditions, such as diabetes, middle ear infection, head injury, high blood pressure, head and neck surgery, stroke, genetic disease, neurological disorders, or respiratory disease were reported by a small number of participants with Bell's palsy. A case-control study conducted in Italy among 381 cases reported no significant difference due to the presence of hypertension or diabetes. However, the likelihood of Bell's palsy increased linearly every year by 2% with age (*Monini et al., 2010*).

Research has recommended allopathic drug therapy following Bell's palsy (*de Almeida et al., 2014*). The use of corticosteroids is recommended to avoid unsatisfactory patient outcomes, and antiviral drug therapy has an additive benefit (*De Almeida et al., 2009*). However, in our study, 64.3% of participants reported not taking the recommended drugs. A review study by *Potterton (2015)* recommended using corticosteroid therapy within 72 h of the onset of Bell's palsy for a better outcome. In the current study physiotherapy and allopathic drug therapy following the onset Bell's palsy among the participants were favored treatment choices. Complementary therapy, such as dry needling, was reported

by 20.5% of participants. A study conducted among the general population ($n = 420$) found only 49.6% favored steroid treatment, while 54.7% favored traditional medicine (*AlYahya, Al-Qernas & Al-Shaheen, 2018*). A study conducted among dental students ($n = 654$) reported that only 39% favored corticosteroid therapy (*Al Meslet et al., 2019*). Traditional Saudi medicine and cauterization (hot iron rod) were used by 18.7% and 13.5% of participants, respectively. This traditional medicine is reported to cause severe burn injuries and complicated wounds (*Aboushanab & AlSanad, 2019*).

### Limitations

This study was a regional study with a relatively small number of participants. Recall bias might have existed because the study required that the participants recall past events. The Qurayyat region of Saudi Arabia is smaller than the other 13 major provinces of Saudi Arabia. Therefore, the results cannot be generalized to the whole nation. The actual number of patients with Bell's palsy in the Qurayyat region of Saudi Arabia could not be determined because some patients might not visit a hospital or choose traditional medicine. This limitation was reduced in our study through regular public advertisements using pamphlets and announcements in public places.

### Clinical implications

The public can be educated about the best treatments available for Bell's palsy to avoid relying only on traditional medicine. Because Bell's palsy is more common in the winter, hospitals and clinics must stock up on medications and increase the number of physicians to ensure positive patient outcomes. The education program should target young adults, as the incidence was found to be high among those aged 21–30 years. To increase drug compliance among at-risk groups, educational programs must emphasize the potential benefits and adverse effects of corticosteroids and antiviral drugs. Due to cultural inhibitions, poor participation from females living in the Qurayyat region is to be expected. A special education program dedicated only to females living in the Qurayyat region may be beneficial.

## CONCLUSIONS

The incidence rate of Bell's palsy was approximately 25.7 per 100,000 per year in the Qurayyat region of Saudi Arabia. Exposure to cold air and influenza were significant risk factors associated with Bell's palsy. Females were predominantly affected by Bell's palsy in the Qurayyat region of Saudi Arabia. Bell's palsy was most common among those aged 21–30 years. The most favored treatment was physiotherapy following Bell's palsy. The population in the Qurayyat region of Saudi Arabia needs to be educated about the potential benefits of combination therapy, rather than relying on complementary or traditional medicine alone, for improved patient outcomes.

## ACKNOWLEDGEMENTS

The authors would like to thank all the patients with Bell's palsy who participated in this study.

### Funding

The authors were supported by the Deanship of Scientific Research at Majmaah University under project number: R-2022-265. The funders had no role in study design, data collection and analysis, decision to publish, or preparation of the manuscript.

### Grant Disclosures

The following grant information was disclosed by the authors:
The Deanship of Scientific Research at Majmaah University under project number: R-2022-265.

### Competing Interests

The authors declare there are no competing interests.

### Author Contributions

- Fahad Alanazi conceived and designed the experiments, performed the experiments, analyzed the data, prepared figures and/or tables, authored or reviewed drafts of the article, and approved the final draft.
- Faizan Z. Kashoo conceived and designed the experiments, performed the experiments, analyzed the data, prepared figures and/or tables, authored or reviewed drafts of the article, and approved the final draft.
- Anas Alduhishy conceived and designed the experiments, performed the experiments, prepared figures and/or tables, authored or reviewed drafts of the article, and approved the final draft.
- Mishal Aldaihan conceived and designed the experiments, performed the experiments, prepared figures and/or tables, and approved the final draft.
- Fuzail Ahmad conceived and designed the experiments, performed the experiments, prepared figures and/or tables, authored or reviewed drafts of the article, and approved the final draft.
- Ahmad Alanazi conceived and designed the experiments, performed the experiments, analyzed the data, prepared figures and/or tables, authored or reviewed drafts of the article, and approved the final draft.

### Human Ethics

The following information was supplied relating to ethical approvals (*i.e.*, approving body and any reference numbers):

Ethical approval was granted by the Ethical comitttee at Qurayyat General Hospital (Ethical Application Ref: QGH-EC-16-2020.

### Data Availability

The raw data is available in the Supplemental File.

## Supplemental Information

Supplemental information for this article can be found online at http://dx.doi.org/10.7717/peerj.14076#supplemental-information.

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
