# Peer review of "Incidence rate, risk factors, and management of Bell’s palsy in the Qurayyat region of Saudi Arabia"

_PeerJ, doi:10.7717/peerj.14076_

## Round 0.1 · original submission · Major Revisions

Dr Alanazi and co-authors. The Reviewers have submitted their comments on your manuscript and there are a number of significant issues you will need to address. Additionally, it would be highly beneficial to contact a professional editor to eview your amended manuscript prior to submitting as the Reviewers have identified a number of grammatical errors. Please address all of the Reviewers' comments in a timely manner. Thanks, A/Prof Mike Climstein

Reviewer 1 ·

Basic reporting

Grammatical and spelling error in places.
Structure reasonable. Literature review appropriate.
- Abstract: Peerj standards- unclear subheadings in abstract are to be bold, but in the manuscript, it is not bold.
- Line 25, 47,132,145,217 a small alphabet “b” has been used for Bell’s palsy on several occasions.
- Line 30-31- framing of sentences needs changed
- Line 33- “exposure” better than exposing as used subsequently.
- Line 35- “years” rather than year.
- Liner 44- “were” instead of was.
- Line 46- “the” instead of “A”
- Line 47, for consistency, use the same term, “physiotherapy” instead of physical therapy.
- Line 50, 51, punctuation.
- Line 51, “on” instead of towards.
- Line 53- no reference, which year?
- Line 54-55: contradicting statement to previous. Described as “common” (line 49) initially and now “relatively rare”.
- Line 66- increases. Also, this sentence appears to be an inference from the previous sentence. Please clarify what you mean by “many neurological disorders” or perhaps remove that phrase?
- Line 75- “some of the” instead of “one of the” as more than one traditional method is mentioned.
- Subheading under materials and methods- to be in bold for better readability.
- Line 105- consisted rather than consisting.
- Line 120- does not make sense. Is it supposed to mean- “the sample size from reviewing medical records was 279?”
- Line 214- could rephrase as recall bias.

Experimental design

Retrospective study.
Though design is relatively clear, inclusion criteria are sometimes unclear, eg. had a cold or covid vaccine "before" Bell's palsy. Duration of "before" not clear. Knowledge gap and research questions identified.
Research methods was appropriate and of high technical standard. However, some areas need clarification.
- Line 62- reference does not mention hepatitis B, instead mentions that some studies implicate Hepatitis C.
- Line 71: “A standard recommended treatment” is stated. However, the reference is a systematic review in Germany. In addition, the treatment regime mentioned in this review is different from what has been stated. Clarification is required on what “standard” and whose “recommendation” are being referred to.
- Reference 19 used in Line 75 is referring to use of Traditional medicine and its use in pain management rather than Bell’s palsy. This may be acceptable as a general reference for traditional medicine rather than its use in Bell’s palsy. It would be helpful to clarify this.
- Line 75-77: Reference 20- excellent one describing traditional method of cauterisation for Bell’s palsy.
- Line 81: suggest “following diagnosis of Bell’s palsy”
- Line 92: Suggest “where a patient was a minor….”
- Line 103- not sure what this phrase is attempting to convey. Suggest remove.
- Questionnaire in appendix 1 which was not there in the document to review.
Line 129-133- Rephrase as does not seem clear. Did the author mean “The incidence was calculated as number of cases of Bell’s palsy per 100,000 population. The total population of the Qurayyat region of Saudi Arabia in the year 2020 was 180,430.” Line 131 does not appear to add much value especially as incidence per 100,000 population is not explicitly stated.
Line 142- Is there are specific duration set for the term “before” with reference to vaccination? The reader has been signposted to table 1 but no reference is made to vaccination in this table.
Line 144- is there a specific duration for “before” in the context of exposure to cold?
Line 158-160: “no significant relationship with consanguinity”. However, 38% participants were related. P value calculation might help establish significance of association.
Line 169- for consistency it is recommended that the same term be used- either physical therapy OR physiotherapy in all places.
Line 179- incidence is referred to in Line 131, but figures not mentioned

Validity of the findings

The author identified a lack of literature in the area prior to research.
Validity of the findings is discussed. However, as stated in the previous section, some statement of "significant" and "non-significant" findings require p value calculation.
Some data is missing- Appendix 1 that is referred to.
Combination treatment is referred to, but no details in tabular format is provided.
Conclusions are well-stated and answers the research questions.

·

Basic reporting

Need improvements.

Experimental design

Need improvements.

Validity of the findings

the findings are valid. However, reporting need improvement.

Additional comments

Thanks for opportunity to review manuscript entitled ‘‘Incidence, risk factors, and management of Bell's palsy in the Qurayyat region of Saudi Arabia’’ for Peerj Journal. The author/authors examined the incidence of Bely’s palsy. The strength of the manuscript includes examining variables of interest in a country where such studies are scarce. Overall, although the article is generally well written and deserves to be published in this journal some necessary and minor revisions must be made before the publication of the article. Because my main philosophy of reviewing a manuscript as reviewer and sometimes an editor to improve the manuscript and not punishing the authors, I provided very specific and detailed peer review of the manuscript to increase its quality and citation potential. I hope authors of the manuscript may benefit from my review. Necessary revisions reported section by section with the page and line number and when possible with suggestions.
Necessary Revisions
General
1.The writing of Bell’s palsy is inconsistent along the manuscript. The correct writing is Bell’s palsy. Author sometimes used correctly and sometimes write wrongly such as Bell’s Palsy or bell's palsy. Authors must correct this mistake along the manuscript.
2. Along the manuscript reporting the statistics are wrong. Authors must correct this problem along the manuscript. I provided some examples as follows. Authors reported the frequencies as (n=12,7.0%). Correct reporting is (n = 12, 7.0%). Another example reporting chi square statistics. Authors reported like this X2(6, N = 171) but correct form is χ2(6, N = 171) = 16.35, p = 0.012. Moreover, authors inconsistently reported statistics along the manuscript sometimes reported two decimal, sometimes reported three decimal to findings. Authors must consistently report consistently two decimal after the dot. Moreover, authors reporting standard is not consistent with Peerj.
3. The in-text citations of manuscript is not consistent with writing rules of Peerj. Authors must correct this along the manuscript.
4. Authors must edit the manuscript from a professional proofreading company after possible revisions. It is very difficult to understand most sentence, and a lot of writing mistakes exist. A lot of sentence used without citations when necessary.
5. Keywords are completely missing in the manuscript and must be added.
Abstract
6. Abstract, Page 6 Line 24-26: Please revise following sentence for clarity. It is very difficult to understand. ‘ ‘A 28-item questionnaire was developed by a team of experts, pre-tested, and piloted among bell's palsy patients before sending it to the eligible participants.’’ Because I am not able to understand this sentence, I have no revision suggestion.
7. Abstract, Page 6, Method Section: Authors must add their used statistical analyses to this section.
8. Abstract, Page 6, Line 29-30: Please revise following sentence ‘ ‘Females (n=141, 86%) were predominantly affected than males.’’ One revision may be that ‘ ‘Bell’s palsy was more commonly occurred in females than males ( n = 141, 86%). Moreover, authors must a sentence to indicate their reached Bell’s palsy patients. When individuals read without this information, they think reported percentages are wrong. Thus, before female sentences author must add this information.
9. Abstract, Page 6, Line 29-30: Please revise following sentence ‘ ‘The most affected age
group was 21-30 years (n=76,44.4%).’’ One revision may be that ‘ ‘Bell’s palsy illness most commonly experienced among 21-30 years old individuals.’’
10. Abstract, Page 6, Line 30-31: Please revise following sentence, it is unclear and difficult to understand. ‘ ‘There were 157 (89.5%) cases affected who reported Bell's palsy for the first time.’’ I think authors want to mean Bell’s palsy illness occurrence number. However, I am not sure. If so, one revision may be that ‘ ‘There were 157 (89.5%) cases who reported Bell’s palsy for the first time.’’
11. Abstract, Page 6, Line 31-32: Please revise following sentence ‘‘Majority of the participants reported right-sided facial paralysis (n=96, 56.1%) and only (n=12,7.0%) had bilateral facial paralysis.’’ One revision may be that ‘ ‘The majority of the participants experienced right-sided facial paralysis (n = 96, 56.1%)’’. No need second part.
12. Abstract, Page 6, Line 33: ‘ ‘Chi-square analysis’’ must be ‘ ‘The chi-square analyses’’
13. Abstract, Page 6, Line 33: ‘ ‘relation’’ must be relationship or association.
14. Abstract, Page 6, Line 35: ‘ ‘The post hoc analysis revealed that’’ must be ‘ ‘The post hoc analyses revealed that’’
15. Abstract, Page 6, Line 35: ‘ ‘20-31-year’’ must be corrected as ‘ ‘ 21-30 year’’.
16. Abstract, Page 6, Line 40: Please remove following ‘ ‘vitamins and neuro-vitality drugs (‘’
17. Abstract, Page 7, Line 41: ‘ ‘The most preferred combination therapy’’ must be corrected as ‘ ‘The most preferred combined therapy’’
17. Abstract, Page 7, Line 46: Please revise following sentence ‘ ‘A peak incidence was seen in the age group. 21-30 years.’’ One revision may be that ‘ ‘ Bell’s palsy most commonly occurred in the age group 21-30 years.’’
Introduction
18. Introduction, Page 7, Line 49: The citation/citations need for following sentence ‘ ‘ Bell’s palsy is a common lower motor nerve paralysis of facial nerve of unknown origin.’’
19. Introduction, Page 7, Line 50-51: The citation/citations need for following sentence ‘ ‘The
patient experiences sudden unilateral flaccid paralysis of muscles of facial expression rarely bilateral.’’
20. Introduction, Page 7, Line 51-52: The citation/citations need for following sentence ‘ ‘The patient is unable to perform facial movements towards the affected side and facial asymmetry becomes clear with attempted facial movements’’
21. Introduction, Page 7, Line 53-54: The citation/citations need for following sentence ‘ ‘The annual incidence is 15-30 per 100,000 populations as per the data from National Health Survey, UK’’
22. Introduction, Page 7, Line 54-55: The citation/citations need for following sentence ‘ ‘Being a relatively rare condition, the annual incidence of Bell's palsy is reported to be 11-40 cases per 100,000 populations.’’
23. Introduction, Page 7, Line 61: Please revise the following sentence ‘‘ The cause of Bell’s palsy is unknown’’ as ‘ ‘The exact cause of Bell’s palsy is unknown’’
24. Introduction, Page 7, Line 61: Please revise the following sentence ‘‘ Introduction, Page 6, Line 61: Please revise the following sentence ‘‘ many probable’’ as ‘ ‘many possible’’
25. Introduction, Page 7, Line 69-70: Please revise the following sentence ‘ ‘Therefore, a study is needed to explore the impact of increased prevalence of risk factors and consanguineous marriage on the prevalence and incidence of Bell’s palsy in Saudi Arabia.’’ In the sentence, three times prevalence used. One revision may be that ‘ ‘Therefore, a study is needed to explore the possible impact of increased risk factors and consanguineous marriage on the incidence of Bell’s palsy in Saudi population.’’
26. Introduction, Page 8, Line 80-83: Please revise the following sentence ‘ ‘Therefore, the study aims to determine the incidence, association of risk factors, and preferred treatment options following Bell’s palsy in the Saudi population.’’ One revision may be that ‘ ‘Therefore, the aim of this study is to determine the incidence, possible risk factors, and preferred treatment options following Bell’s palsy in the Saudi population.’’.
Method
27. Method section, General: Author must rearrange Methos section using the subtitles Research Design, Participant, Data Collection Tools, Procedure, and Statistical Analyses. In this form it looks very messy.
28. Method, Page 8, Line 116-117: As an expert in multivariate analyst, I can confidently say that the statement ‘ ‘The internal consistency of the 28 item questionnaire was 0.716 calculated by Cronbach's alpha.’’ Was wrong and must remove from the manuscript.
29. Method, Page 9, Line 134-135: As an expert in multivariate analyst, I can confidently say that the statement ‘ ‘The questionnaire was analyzed for internal consistency by Cronbach alpha and Inter-Class correlation coefficient.’’ was wrong and must remove from the manuscript.
30. Method, Statistical Analyses section: Authors must add used significance level in this section. Moreover, are the authors examined the assumptions of chi square analyses? I observed a lot of class below 5 participants and strong concerns about the assumption violations. If authors examined and satisfied the assumption of these analyses, they must add a sentence to statistical analyses section to indicate this. If assumptions are violated authors must use in these analyses Fisher’s exact test.
Results
31. Results, Page 9, Line 158-159: Please revise the following sentence ‘ ‘The study consists of a majority of female participants (n=147, 86%) out of 171 total sample sizes.’’, One revision may be that ‘ ‘Among 171 Bell’s palsy patients, the majority of participants were female (n = 147, 86%).’’
Discussion
32. Discussion, Page 10 Line 162-163: Please revise the following sentence ‘ ‘The study aimed to identify the risk factors and preferred treatment after Bell's palsy among participants residing in the Qurayyat region of Saudi Arabia.’’ The correct version is ‘ ‘Theis study aimed to examine incidence of Bell’s palsy and identify the risk factors and preferred treatment after this illness among participants residing in the Qurayyat region of Saudi Arabia.’’
33. Discussion, General: the practical implications of study findings are completely missing in the manuscript. Authors must add a section named Practical implications and must discuss practical implications of their study findings.
34. Discussion, Page 11 Line 225: Please revise following sentence ‘ ‘A peak incidence was seen in the age group. 21-30 years.’’ One revision may be that ‘ ‘ Bell’s palsy most commonly occurred in the age group 21-30 years.’’
Tables and Figures.
35. In Table 1 age variable ‘‘60 above’’ must be ‘‘61 or above’’
36. In Table 1, Recurrence variable ‘ ‘Frist time’’ must be ‘‘first time’’
37. In Table 1, following changes must be made and analyses must be repeated and reported after this changes combine 61 or above group with 51-60 and rename this group as 51 or above. Moreover, Recurrence variable must be two group first time second or more For Treatment following Bell's Palsy Vitamins and nerve supplements, No treatment, Surgery must remove from analyses or combined as Other.
38. Table 2 is very complicated author must simplify this table.
39. Risk factors in Figure 2 for this study or compilation from other study. If compiled from other studies citation need for this.

---

## Round 0.2 · Major Revisions

Dr Kasgoo. I have received the 2 reviews of your manuscript from the reviewers and there are major changes recommended. May I recommend that you and your co-authors revise your manuscript according to the reviewers' recommendations and ensure you address all of their concerns.

Again, I recommend you have a professional editor review your amended manuscript prior to resubmitting. Thanks, A/Prof Mike Climstein

Reviewer 1 ·

Basic reporting

Still has some grammatical errors. Structure is good.
- This time in correctly written as Bell’s Palsy. The word “palsy” is a common noun and is to be in small letters unless at the start of a sentence as it is a common noun.
- Abstract- refers to Bell’s palsy as “temporary”, not sure why this was changed from before. This is not always the case.
- In text citation still does not comply with journal guidelines. – journal guideline is author(s) with date of publicatrion.
- Line 124- grammatical error- should read- indicated as unclear….
- Line 150-151- please rephrase sentence with n= 21 in brackets.
Line 158/159- again, n= 20 should be in brackets. Should read- twenty participants (n=20) (11.7%) reported having a
- 159 familial-related genetic disorder.
- 160- recurrence
- Line 168-169 and 176-178 contradicting each other

Experimental design

Research question is clear.
Questionnaire aims to answer the research questions.
High technical, ethical and academic standard of approach.
Findings discussed in lines 195- 201 are full of contradicting statements.

Validity of the findings

Findings restricted to a particular region.
conclusion clear

Additional comments

Significant improvement since original review but stil needs further amendments to reach the standard of publication

·

Basic reporting

Basic reporting need significant improvements.

Experimental design

Experimental design needs significant improvements.

Validity of the findings

I have some concerns. See my comments.

Additional comments

Thanks for opportunity to review revised manuscript entitled ‘‘Incidence, risk factors, and management of Bell's palsy in the Qurayyat region of Saudi Arabia’’ for Peerj Journal. The author/authors examined the incidence of Bely’s palsy. Although authors slightly improved the manuscript from the first review the article, the article still requires significant improvements in almost all sections. Most of revisions also not adequate and some of them are wrong. Overall, as an experienced article editor and reviewer, I think this article still not suitable for publication in this journal and requires a major revision.
General
1. Along the manuscript reporting the statistics are wrong. Authors must correct this problem along the manuscript. I provided some examples as follows. Authors reported the frequencies as (n=12,7.0%). Correct reporting is (n = 12, 7.0%). Another example reporting chi square statistics. Authors reported like this χ2 (6, N = 171) but correct form is χ2(6, N = 171) = 16.35, p = 0.012. Moreover, authors inconsistently reported statistics along the manuscript sometimes reported two decimal, sometimes reported three decimal to findings. Authors must consistently report consistently two decimal after the dot except for p value. Moreover, authors reporting standard is not consistent with Peerj.
2. The in-text citations of manuscript is not consistent with writing rules of Peerj. Authors must correct this along the manuscript. For example, (Greco et al.) must be (Greco et al., 2012). Another example is (Colella et al., ) (Potterton) and must be (Potterton, 2015; Colella et al., 2021). All in text citations are wrong and must be corrected.
3. Authors must edit the manuscript from a professional proofreading company after possible revisions. It is very difficult to understand most sentence, and a lot of writing mistakes exist. A lot of sentence used without citations when necessary.
4. Title must be revised as follow: Incidence rate, risk factors, and management of Bell's palsy in the Qurayyat region of Saudi Arabia’’
5. I think all incidence statements must revise as incidence rates. I looked the definition of incidence rate and this is exactly what authors to do.
The term incidence rate refers to the rate at which a new event occurs over a specified period of time. Put simply, the incidence rate is the number of new cases within a time period (the numerator) as a proportion of the number of people at risk for the disease (the denominator).
https://www.cdc.gov/STD/Sassi/Module2/how_to_calculate_incidence_rate.html
New Necessary Revisions
6. Abstract, Line 26: Data is a plural noun. Thus, ‘‘The data was……….’’ must be ‘ ‘Data were…..’’
7. Abstract, Line 31: (n=147, 86.0%) in this n must be italic add a space before and after =
8. Abstract, Line 33: (n = 76, 44.4%). in this n must be italic
9. Abstract, Line 34-35: (n = 96, 56.1%) in this n must be italic add a space before and after =
10. Abstract, Line 36-37: (χ2(6, N = 171) = 14.926, P = 0.021), χ2(6, N = 171) = 16.354, P= 0.012, this must be corrected as (χ2(6, N = 171) = 14.93, p = 0.021), χ2(6, N = 171) = 16.35, p = 0.012
11. Abstract, Line 40: (n=149, 87.1 %) in this n must be italic add a space before and after =
12. Abstract, Line 41: (n=61, 35.7%) in this n must be italic add a space before and after =
13. Abstract, Line 41: (n=35, 20.5%) in this n must be italic add a space before and after =
14. Abstract, Line 42: (n=32, 18.7%) in this n must be italic add a space before and after =
15. Abstract, Line 43: (n=2, 1.2%) in this n must be italic add a space before and after =
16. Abstract, Keywords: Authors must use comma between the keywords. They must not use ;
17. Introduction, Line 62: Please check and correct citation JI et al.
18. Introduction, Line 67: 27.8 % must be 27.8%
19. Introduction, Line 67: 26.3 % must be 26.3%
20. Method, Line 96: Study design and settings: must report as subtitle not like this. The same thing also valid for Study Method: For example,
Study design and settings
The study ……….
21. Method, Line 96: Please revise following sentence ‘ ‘The study is a retrospective cross-sectional hospital-based study.’’ One revision may be that ‘ ‘This research was a retrospective cross-sectional hospital-based study.’’
22. Method, Line 107: Study Method section must rename as Procedure.
23. Method, Line 122: Authors added a questionnaire nut it is in Arabic language and not possible to control it. Author must upload questionnaire in English. Moreover, in this section authors must give more information about content of questionnaire. This section gives nothing about content of questionnaire. Authors specifically, give information about asked questions in this sections with response options. Moreover following statement ‘ ‘The internal consistency of the 28 item questionnaire was 0.716 calculated by Cronbach's alpha.’’ Was wrong and must remove from manuscript. Authors can only calculate if questionnaire content aim the measure the same underlying construct.
24. Method, Line 130: The sample size calculation title must be revised as Population and Sample. Moreover, authors must add following information to this section. At worst, authors must add, The gender distribution of sample, and the mean, standard deviation, minimum and maximum values of age.
25. Method, General: Authors must rearrange Method section with subtitles as follows with the same order. Study design and settings, Population and Sample, Questionnaire, Procedure, and Statistical Analyses.
26. Method, Line 141: What is abbreviation of MOH?, This is the first use and author must provide long name of this.
27. Method, Line 144-145: Authors must narratively add incidence formula or must cite a researcher for formula. For example, Incidence formula proposed by ……used to calculate incidence rate in Qurayyat region of Saudi Arabia.
26. Results, Line 147: (n = 147, 86%) in this n must be italic.
27. Results, Line 148: (n=76,44.4%) in this n must be italic add a space before and after =
28. Results, Line 149: (n=11,6.4%) in this n must be italic add a space before and after =
29. Results, Line 150: (n=129, 75.45%) in this n must be italic add a space before and after =
30. Results, Line 150: (n=18, 10.5%) in this n must be italic add a space before and after =
31. Results, Line 151: Following sentences must be corrected ‘ ‘There were n=21 (12.3%) participants vaccinated before experiencing Bell’s Palsy.’’ The correct version is ‘ ‘There were 21 (12.3%) participants vaccinated before experiencing Bell’s Palsy.’’
32. Results, Line 151: (Table 1). Must be after the sentence ‘ ‘There was a significant number of participants (n=135,78.9%) exposed to cold air before experiencing Bell’s Palsy.’’
33. Results, Line 155: χ2 (5, N = 171) = 14.926, P = 0.011, this must be corrected as χ2 (5, N = 171) = 14.93, p = 0.011
34. Results, Line 156-157: the following information in the text ‘ ‘The post hoc analysis with Bonferroni correction and adjusted P-value of 0.0072 to be significant at the P<0.05 level, revealed 21-30 year age group was significantly affected.’’ and Table 1 is not consistent with each other.
35. Results, Line 158: and n=20 must be revised as 20
36. Results, Line 157: P-value of 0.0072 must be p-value of 0.007
37. Results, Line 157: P<0.05 level must be p < 0.05 level
38. Results, General: Statistical analyses reported in the text and in the Table is not consistent with each other. Authors must carefully check both of them and correct this. Moreover, in the Table 1 analysis of COVİD vaccination is missing. Moreover, analyses regarding to Related Samples Cochran’s Q Test is wrong. It is used in repeated measures design and must be corrected.
39. Results, General: Statistical symbols in the following sentences must be corrected as indicaten in above. Please look comment 26 to comment 33.
consanguinity with gender, onset and reoccurrence of Bell’s Palsy but consanguinity showed
significant relationship with side affected and age group, χ2 (2, N = 171) = 12.090, P = 0.002, χ2 (5, N = 171) = 13.025, P = 0.023 respectively, The main therapeutic approach preferred was physiotherapy (n=149, 87.1 %), followed by corticosteroids and antivirals drugs (n=61, 35.7%), acupressure (n=35, 20.5%), traditional Saudi herb medicine (n=32, 18.7%), cauterization by hot iron rod (n=23, 13.5%), supplementary therapy (vitamins and neuro-vitality drugs (n=2, 1.2%), facial cosmetic surgery (n=1,0.6%) and no treatment (n=1,0.6%). The most preferred combination therapy was physiotherapy (87.6%) with corticosteroid and antiviral drugs (35.9%), and acupressure (17.6%) (Table 2) There were relatively less number of participants suffering from ear infection (n=28, 16.4%), diabetes (n=23,13.5%), genetic disease (n=20, 11.7%), high blood pressure (n=18, 10.5%), neurological disorder (n=16, 9.4%), head injury (n=11, 6.4%), balance problem (n=10, 5.8%) stroke (n=3, 1.8%), and heart disease (n=3, 1.8%) (Figure 2).
40. Clinical Implications section must significantly improve. Authors must add at least three to four sentences in this section.

---

## Round 0.3 · Minor Revisions

Dr Alanazi, the Reviewer has requested further, minor changes to your manuscript. Can you please make these changes at your earliest convenience and resubmit. Thanks, A/Prof Mike Climstein

·

Basic reporting

Clear and unambiguous, professional English used throughout.

Experimental design

Rigorous investigation performed to a high technical & ethical standard.

Validity of the findings

The findings are valid.

Additional comments

Thanks for opportunity review revised manuscript entitled ‘‘Incidence rate, risk factors, and management of Bell’s palsy in the Qurayyat region of Saudi Arabia’’. I would like the thanks to authors. They make a good job for improving quality of their manuscript. Authors revised the manuscript as I requested with a good will. However,some minor hings left behind. After this corrections I recommend accept decision.
1. Please correct writing of chi square along the manuscript. You can search in word Greek small letter chi and then add upper 2.
2. Please re upload latest versions of all tables, without corrections (They must not be track changes or red underlined).
3. Please do all references the same letter size with manuscript body.

---

## Round 0.4 · accepted · Accept

Dr Kashoo, thank you and your colleagues for making the changes to your manuscript. I am pleased to recommend your manuscript for publication to the Editor. Thank you again for choosing PeerJ, we sincerely appreciate your support and look forward to receiving future submissions. Thanks, A/Prof Mike Climstein